# Chemical Variability and Biological Potential of *Cornu aspersum* Mucus as a Source for the Development of New Cosmetic and Pharmaceutical Products

**DOI:** 10.3390/molecules30214197

**Published:** 2025-10-27

**Authors:** Nataliia Hudz, Vira Turkina, Tetyana Alyokhina, Olena Motyka, Nataliia Chemerys, Tetiana Rumynska, Andrii Lozynskyi, Izabela Jasicka-Misiak, Lesya Kobylinska

**Affiliations:** 1Department of Pharmacy and Ecological Chemistry, Institute of Chemistry, University of Opole, 45-040 Opole, Poland; izabela.jasicka@uni.opole.pl; 2Department of Drug Technology and Biopharmacy, Danylo Halytsky Lviv National Medical University, Pekarska Street 69, 79010 Lviv, Ukraine; 3Department of Biological Chemistry, Danylo Halytsky Lviv National Medical University, Pekarska Street 69, 79010 Lviv, Ukraine; ver.apachi85@gmail.com (V.T.); alyokhinalvov@gmail.com (T.A.); kobylinska_lesya@meduniv.lviv.ua (L.K.); 4Research Institute of Epidemiology and Hygiene, Danylo Halytsky Lviv National Medical University, Pekarska Street 69, 79010 Lviv, Ukraine; difteriandi@gmail.com; 5Department of Psychiatry and Child Psychiatry, Psychotherapy and Clinical Psychology, Danylo Halytsky Lviv National Medical University, Pekarska Street 69, 79010 Lviv, Ukraine; chemerus.talj@gmail.com; 6Department of Microbiology, Danylo Halytsky Lviv National Medical University, Pekarska Street 69, 79010 Lviv, Ukraine; tanityshka.r@ukr.net; 7Department of Pharmaceutical, Organic and Bioorganic Chemistry, Danylo Halytsky Lviv National Medical University, Pekarska Street 69, 79010 Lviv, Ukraine; lozynskyiandrii@gmail.com

**Keywords:** *Cornu aspersum* mucus, mucin, snail secretions, cosmetology, wounds

## Abstract

Snail mucin is one of the animal products widely used in cosmetic products. The mucus of *Cornu aspersum* (*C. aspersum*) contains compounds that have antibacterial, antioxidant, proliferative, pro-migration, angiogenesis-promoting, and other biological effects. This study aimed to critically analyze and consolidate existing data on the bioactive components of *C. aspersum* mucus and the mechanisms of their influence on human health, focusing mainly on its cosmetic, regenerative, anti-inflammatory, and antimicrobial properties. We conducted a literature search analysis on this problem using the following search databases in English: PubMed, PubChem, Mendeley, Google Scholar, Scirus, DOAJ, BASE, CORE, Science.gov, and RefSeek up to 12 August 2025. It was shown that snail mucus facilitates wound healing, which could be the prerequisite for the development of innovative formulations for the adjuvant therapy of skin wounds. However, there are problems with the standardization of snail mucus because of the absence of single quality indexes, their limits, and the complicated structure of snail mucins. Moreover, there is a lack of clinical randomized trials evaluating the safety and efficacy of *C. aspersum* snail mucus. In conclusion, snail mucus’s biological effects deserve further investigation and pave the way for further studies of its potential as a raw material for pharmaceutical products, including the chemical structure of the still unknown molecules, its standardization, nonclinical and clinical studies, and further studies of snail mucus for its usage in cosmetology.

## 1. Introduction

Scientists in the fields of medicine and cosmetology have been actively pursuing new natural compounds and products with biological properties for several reasons. Among them are consumers who are more image-conscious and concerned with skin health and beauty, the globalization of the beauty market, a paradigm shift to the consolidation of green chemistry, the introduction into cosmetic products of unconventional ingredients of animal origin in addition to herbal extracts for enhancing the biological effects of these products, an increase in the resistance of many microorganisms to known antimicrobial drug substances, etc. [1,2,3,4,5,6].

In the 1990s, a paradigm shift among chemists started, contributing to the consolidation of green chemistry as a legitimate scientific sphere [1]. On the one hand, this was facilitated by the active popularization of the information by social media and Internet users about the risks associated with the use of multiple chemical compounds in cosmetology and the benefits of natural products [1,2]. On the other hand, there has been a continuous widening of the scientific knowledge about the chemical composition of natural products and the mechanisms of their effects on the skin and body due to the development of modern analytical methods and conducting studies aimed at searching for and justifying the properties of snail mucus [2,3,4,5,6]. And last but not least, studies of snail mucus are topical for medicine, considering an increase in the number of bacterial strains resistant to conventional antimicrobial drug substances, which threatens public health safety, and the treatment of wounds [7,8,9].

Snails and slugs are gastropods in the phylum *Mollusca*, which are found in many habitats, including forests, freshwater ponds, rivers, oceans, gardens, and mangroves. Gastropods produce mucus in epithelial glands covering their body [10]. The common garden snail (*C. aspersum*), formerly known as *Cryptomphalus aspersa*/*Helix aspersa* (*H. aspersa*), is an invasive land snail and agricultural pest of global significance, which has spread from its native Europe to most continents. Its ability to adapt to a broad variety of environments can be attributed to several factors [11].

The idea of using mucus or snail secretion dates back to ancient times [12,13]. Unconventional ingredients of animal origin, such as snail mucin, propolis, starfish powder, and bee venom, in addition to herbal extracts, are added to cosmetic and pharmaceutical products to enhance their biological effects [12,13,14,15]. Different compounds with multifunctional biological effects (antibacterial, antitumor, antioxidant, reparative, etc.) in the *C. aspersum* mucus are supposed to determine the potential of its utilization in innovative cosmetic and pharmaceutical formulations [8,9,12,13,14,16,17]. Moreover, this raw material is readily available due to the network of snail farms, which have been actively developing in recent years in Ukraine, Bulgaria, China, Indonesia, Romania, Spain, Morocco, etc. [18,19,20]. However, the field of snail mucus research remains surprisingly underdeveloped for many reasons [13], some of which are considered in this paper.

Invertebrates do not have an adaptive (acquired) immune system [5,8]. They only have an innate immune system for the battle against invading diseases. For this reason, their innate immune systems can be quite powerful. This finding has brought about numerous studies about several invertebrate species in recent years, including properties of snail mucus and secondary metabolites of snails [5,8,9,21,22,23,24,25]. Moreover, some species *H. pomatia*, *H. aspersa maxima*, and *H. aspersa* (Müller, 1774) are used as food. It is worth noting that Helicidine^®^, an extract from *H. pomatia* L., was authorized by the French National Authority for Health as an antitussive medicinal product. Helix Med (snail mucus) and Alsiroyal (snail syrup) are commercialized for the treatment of wet and dry cough and fluidification of airway mucus [25].

The purpose of this study was to critically analyze and summarize the available data on biologically active components of the *C. aspersum* mucus, the mechanisms of its effect on cells, the human skin, and mucous membranes, which could determine its role in pharmaceutical formulations of anti-inflammatory and wound-healing activity and cosmetic products.

The study embraces a literature review based on exploratory papers, research, critical analysis, and interpretations of results of the studies described in the scientific papers. We conducted a critical literature analysis on this problem using the following databases: PubMed, PubChem, Mendeley, Google Scholar, Scirus, DOAJ, BASE, CORE, Science.gov, and RefSeek. The search was performed in an advanced form. We used the following keywords: snail secretions, mucin, mucus, proteoglycans, and glycosaminoglycans. The criterion of the inclusion of the scientific publications referred to the language, because we analyzed publications in English. There was no exclusion criterion for the period of publication of a paper. This review included laboratory studies in vitro and in vivo and clinical trials. Some authors (V.T., T.R., O.M., T.A., N.C., and N.H.) independently evaluated titles and abstracts, followed by full-text assessments based on predefined eligibility criteria and outlined. Discrepancies were resolved through discussion, and unresolved issues were adjudicated by other authors (L.K., A.L., and N.H.). Critical analysis was mainly performed by N.H.

In the future, the performed survey will allow substantiating the targeted use of snail mucus in the development of new, effective, and safe cosmetic and pharmaceutical products or medical devices.

## 2. Chemical Composition of the *C. aspersum* Mucus

Qualitative and quantitative characteristics of mucus depend on snails’ housing conditions, the season, the diet, the collection method, the species of snail, the methods of mucus preservation (freezing, sublimation, etc.), the type of secretory glands and tissue structures, etc. [8,10,13,17,23,25]. About 50% of the core peptides may be shared through their evolutionary relationship and are essential to mucus function; namely, they are core proteins of mucins and other proteoglycans produced by different gastropods [10]. Secreted mucins are a family of heavily glycosylated proteins produced in epithelial cells in most animals and humans [13,26]. These mucins are responsible for the biological properties of mucus [13,22]. The advanced analytical methods allowed deepening the knowledge about *C. aspersum* mucus chemical composition and the robustness of indexes. Among the analytical methods are infrared spectrometry (IR), HPLC, spectrophotometric analysis with Folin–Ciocalteu reagent for the evaluation of the protein quality, the allantoin and glycolic acid content, polyphenols, respectively, MALDI-TOF-TOF mass spectrometry for the determination of molecular mass of fractions of snail mucus, MALDI-MS/MS analysis for the determination of the amino acid sequences of peptides, etc. [5,8,22,25,27]. It is believed that the active substances of mucus make it a unique natural product that cannot be reproduced in a laboratory by chemical synthesis [27,28].

Snail secretion consists of a diluted network of polymers and low molecular substances [5,8,22,24,25,29]. Mucous secretion is industrially treated before being added to creams, cleansers, serums, and other forms [24,27]. Unexpectedly, Cristiano & Guagni classify snail mucus as a product derived from insects, along with, for example, propolis derived from bees [24]. Despite what has just been said concerning the classification, snail mucus is among the ingredients obtained without causing suffering or harm to snails using the non-lethal Muller method (acid stimulation) or electrical stimulation at low voltage; therefore, there are no ethical concerns with the industrial manufacture of snail mucus using these methods [17,24,28]. Unfortunately, there are some lethal methods, such as crushing whole snails, using solvent extraction, mechanical stimulation, or centrifugation, which raise large ethical concerns [17].

It was also found that mucus contained either acidic glycoproteins, calcium salts, proteins, or fat globules and flavones, depending on the gland localization and, accordingly, the cell type producing mucus. The modern publications identify four types of secretory glands in *C. aspersum*. The first two produce acidic mucopolysaccharides, the third type produces calcium compounds, and the fourth produces proteins [10]. The mucus facilitates snail feeding, locomotion on different surfaces, skin protection, hydration, adhesion, and defense against infection and predators, or even to attract mates for reproduction [10,11,27].

The IR analysis revealed some absorbance peaks, which bring important general information about the composition of the *C. aspersum* mucus. The absorbance peak at 3250 cm^−1^ was related to hydroxylic groups of hydrophilic amino acids. The area between 3000 and 3200 cm^−1^ is defined by aromatic overtone due to aromatic amino acids. The peaks at 1645 cm^−1^ and at 1540 cm^−1^ are related to amide bonds that indicate the presence of proteins [27]. Similar results of the IR analysis were obtained by the Polish scientists [30]. According to X-ray spectral analysis, the mineral composition of the ash residue of the lyophilized *C. aspersum* mucus contained Cl (64.7%), Ca (18.1%), K (6.5%), S (5.7%), P (2.6%), and Mg (1.7%). The number of proteins in the water-soluble fraction, measured with a biuret test, is 24.33 g/L [6]. According to Cerullo et al. (2023), the protein concentration in the resuspended crude mucus solution reached about 77–170 g/L [31]. In general, the secreted mucus is a heterogeneous hydrogel that contains a large fraction of high molecular weight, highly glycosylated mucin proteins (glycoproteins). The garden snail mucus consists of 90–99% water and glycosaminoglycans (hyaluronic acid), allantoin, metalloproteins containing copper, zinc, iron, and manganese, glycoproteins (mucin, lectin, and mitomycin), polyphenols, various enzymes, peptides, proteins, different vitamins, glycolic acid, etc. [5,6,17,22,25,27,28,30,32,33,34,35,36].

Antioxidant enzymes are critical in the neutralization of reactive oxygen species (ROS) and mitigation of oxidative damage [32,33]. Such antimicrobial proteins with an MW > 20 kDa, like aspernin, hemocyanins, H-lectins, L-amino acid oxidase-like protein, and mucins (mucin-5AC, mucin-5B, mucin-2, and mucin-17), were determined via a proteomic analysis on sodium dodecyl sulfate–polyacrylamide gel electrophoresis and bioinformatics in the mucus of *C. aspersum* [8]. According to the USA Food and Drug Administration, allantoin is a safe and effective active substance for skin protection at a concentration of 0.5% to 2.0% [35]. Glycolic acid, allantoin, collagen, and elastin play an important role in gastric mucosa protection and homeostasis [29].

## 3. Aspects of the Standardization of Snail Mucus from *H. aspersa*

Concerning the standardization of snail mucus from *H. aspersa*, it is worth paying attention to the fact that different authors [27,28,29,33,37] provide somewhat or even quite different criteria of acceptability (Table 1). To our mind, such discrepancies in the chemical composition do not facilitate the introduction of snail mucus into medical practice because a therapeutic effect depends on the concentration of an active substance in a medicinal product.

Some scientists standardize snail mucus by the total phenolic content spectrophotometrically. The powdered snail mucus extract was standardized to contain ≥0.3% polyphenols [36]. However, this quality index also has different limits for different authors. The characterization of the biological properties of mucus and the development of medicinal products require the standardization of the mucus. It is worth noting that snail mucus, like propolis, the other preparation of animal origin, is hard to standardize [15,25]. The absence of a single criterion of standardization creates significant obstacles for the development of medicinal products on the basis of snail mucus. Moreover, the EFSA Panel on Nutrition, Novel Foods, and Food Allergens could not establish the safety of the product of snail mucus as a novel food because of the concerns with robustness and consistency of the data submitted by the applicant and their appropriateness for a safety assessment [25]. This paper [25] is significant proof that standardization poses a serious challenge to the implementation of snail mucus into the pharmaceutical industry. For instance, the content of glycolic acid ranged from 0.068 to 0.107 g/L, and the content of glycosaminoglycan sulfate ranged from 0.039 to 0.082 g/L [25]. Such limits are too wide for the pharmaceutical industry. Moreover, it was considered that acute and subacute peroral toxicity studies in CD-1 mice were not performed according to the GLP and OECD guidelines [25]. And last but not least, snail microbiota is another aspect of *C. aspersum* mucus quality. Special aspects of garden snail breeding and maintenance determine the potential threat of their mucus microbiological quality [19]. Microbiological tests using differential diagnostic agar media confirmed that many saprophytic bacteria were found in the snail mucus samples, including *Escherichia coli* and *Staphylococci* bacteria in some samples [5]. The extracted secretion should be prefiltered and then microfiltered in a sterile environment on 0.22-μm membranes to ensure microbiological purity as an essential quality index of all sterile products [25,27,29], because snail mucus is too rich in bacteria (10.78 × 10^10^ CFU/mL) [5]. As precautionary measures, it is suggested to use preservatives and fine filtration using acetyl cellulose fine-pored 2 μm filters [19,29]. Microfiltration reduces microbial contamination of snail mucus [17] and ensures the quality and safety of snail mucus as a raw material for cosmetic and pharmaceutical products. Symbiotic bacteria have an impact on the synthesis and degradation of mollusks’ active substances, maintaining homeostasis, creating a barrier against pathogenic bacteria-caused infections, and inhibiting pathogenic microbial flora. In addition, sequencing showed that the mucus microbiome of *C. aspersum* has unique characteristics, and its representatives belonged to the *Acinetobacter* genus, *Rhizobiaceae* family, *Pedobacter*, *Aeromonas*, *Shewanella*, and *Alcaligenes* genera [5].

The complicated structure of snail mucus arouses a lot of attention and is one more problem in the standardization of snail mucus, considering that the mucin-type glycosylation determines biological functions of mucin. Such 82, 97, and 175 kDa fractions were identified in *C. aspersum* mucus proteins [10]. Ballard et al.’s (2021) [10,11] research is interesting in terms of evaluating the protein composition of snail mucus. Using mass spectrometric analysis, the researchers identified 175 proteins in the mucus traces left by snails during movement, twenty-nine of which had no analogs in the database of unique protein sequences. Of the detected mucus proteins, twenty-two correspond to secretory proteins, including a perlucin-like protein. Many mucus trace proteins are annotated with well-known structural proteins, including collagen and actin, as well as hemocyanin, which is capable of binding oxygen. The gas chromatography method revealed some identified and non-identified volatile compounds, including propanoic acid, benzaldehyde, and limonene [11]. NMR spectroscopy was used to study metabolic profiles of low molecular weight fractions (<1 kDa and <3 kDa) of lyophilized *C. aspersum* mucus. Free primary amino acids were identified in all the fractions; sugars (glucose and sucrose); intermediate metabolites (fumarate); osmolytes (betaine and choline); and several organic acids, including isovaleric, lactic, tartaric, and acetic, as well as allantoin, glutathione, and antimicrobial peptides [22,38].

The thing that is most important is that mucin-type glycosylation can regulate various cellular functions [39]. Moreover, each snail species secretes multiple distinct functional mucus. It is worth paying special attention to modern ideas about mucin structure and its biological functions. A mucin molecule consists of a long peptide chain (apomucin) and several hundred oligosaccharide chains attached to it. In the center of the protein backbone, there is a tandem repeat domain rich in serine, threonine, and proline, which is called the PTS (proline, threonine, serine) or VNTR (Variable Number Tandem Repeat) region. This domain serves as an anchor for glycosylation. The PTS domain is highly O-glycosylated at the threonine and serine residues. The mucin adopts an extended “bottle brush” conformation. Rich in cysteine sites, it forms disulfide bonds [13,26,39,40]. The length of the glycosylation domain and the number of repeats are individual for different mucins and determine their chemical and biological properties. Trace amounts of sialic acids were found in snail mucus. Minor changes in mucin structure, particularly in amino acid sequence and glycosylation, may correspond to completely different biological functions. Glycoproteins with a mucin domain define an extracellular microenvironment with a high density and molecular weight of their associated glycans (up to 80–90 percent of the total glycoprotein mass), which form the main component of glycocalyx. The mucin-dominant glycocalyx is believed to act as a barrier against pathogens [13,39].

Peptides in *C. aspersum* mucous secretions are characterized by excessive amounts of Ser/Thr residues, *N*-acetylgalactosamine, galactose, and fucose glycans. Their average molecular weight is 30 kDa, unlike mammalian mucins, which usually have a larger mass ranging from 100 kDa to 1 MDa with a large amount of sialic acids. In addition, mollusk mucus contains proteins and glycans that are not found in mammalian mucus [31].

Due to the complex chemical composition of *C. aspersum*, at this stage, it is impossible to understand which chemical substances are more responsible for biological properties. There is data on comparative studies of *C. aspersum* mucus effect compared to glycolic acid, which is one of its main components. The mucus showed a higher proliferative capacity compared to glycolic acid, used as a positive control. The fibroblast proliferation induced by the Helix Complex was comparable to, or even higher than, that induced by glycolic acid in terms of fibroblast proliferation [27]. For this reason, Trapella et al. (2018) speculated that the beneficial effect of snail mucus (inducing proliferation and migration of fibroblasts) is a result of all the molecules present in the mixture, not only allantoin and glycolic acid [27].

Snail mucus has also gained interest in the development of novel naturally derived nanomaterials due to improved biocompatibility and biodegradability [36].

## 4. Biological Properties of Some Components of Snail Mucus

The data of analytical studies on snail mucus indicates the presence of biologically active compounds in its composition, which have biological properties and are already widely used in medicinal and cosmetic products and medicinal devices. Table 2 shows the properties of the components of snail mucus.

## 5. *C. aspersum* Mucus Application in Cosmetology

The snail secretion filtrate is the most common raw material in cosmetics [2,32]. Market research findings indicate a growing commercial interest in *C. aspersum* mucus as a raw material for cosmetics. It is expected that the global market for cosmetic products containing snail mucus may reach $982.7 million in 2031 with an average annual growth rate of 8.3% [48]. In the European Union, snail secretion filtrate (INCI: Snail Secretion Filtrate) and fermented snail secretion/mucus filtrate (INCI: Saccharomyces/Snail Secretion) are used as cosmetic raw materials, which are included in the Cosmetic Ingredients Database CosIng [49].

Cosmetic products in different forms containing snail mucus (ampoules, creams, serums, etc.) are being developed that are designed to decrease stretch marks, acne, and pigment spots; to soothe and moisturize the skin; to eliminate signs of photoaging and reduce damage caused by free radicals; to fight age-related skin changes; to recover gray hair color; etc. [16,28,32,50,51,52,53,54,55,56]. At the same time, there are a few scientific publications that justify in a well-argued manner the effectiveness of such cosmetic products. We critically analyzed the biological effects of some preparations with snail mucus for care of the skin, including nutricosmetics [32,54,55,56,57,58,59,60,61,62,63].

For the study of the influence of preparations, including those with snail mucus, or different procedures on the skin, researchers use different methods and biochemical indicators. Among them are numerous biomarkers (Ki67 and p53 as proliferation markers, COX-2 as an inflammation marker) or indices, including corneometric levels, the measurement of the skin elasticity, transepidermal water loss (TEWL), dermoscopic, histological, morphometric, and histopathological evaluations, etc. [32,53,54,55,56,60,64]. The parameter of TEWL evaluates skin barrier function by means of measuring the amount of water that evaporates through the skin. Decreased TEWL can be an indicator for restoring barrier functions, while increased TEWL can be related to the disruption of the epidermal barrier [56].

The important components of the extracellular matrix of skin are collagen and elastic fibers, which degrade under the influence of different factors, among which is ultraviolet (UV) radiation, forming wrinkles, increased skin laxity, and loss of skin firmness [32,57,60]. On the one hand, extracellular matrix metalloproteinases (MMPs) are responsible for the reorganization of the extracellular matrix of skin, including the arrangement and structure of collagen fibers, wound healing, and scarring. On the other hand, MMPs are involved in the development of some skin diseases. Photoaging of the skin may be associated with changes in the expression of MMP-2 (gelatinases). The balance between MMPs and their inhibitors is extremely important for the normal function and morphology of organs, including skin [32]. An increased expression of collagenases (MMP-1) results in degradation of collagen fibers, whereas an increased activity of elastases (MMP-2 and MMP-9) leads to elastin degradation [57]. After the UV irradiation, the moisture content in the skin decreases significantly, and UV-induced ROS generation increases MMP-1 expression in human keratinocytes [57,60].

The Polish scientists studied the influence of ampoules with snail mucus on skin after mesotherapy [32]. The study was carried out during the autumn–winter period. Eighteen women underwent a series of six mesotherapy treatments at intervals of 14 days. They had been randomly divided into three groups (A, B, or C), and there were six women in each group. Groups A and C had micro-needle mesotherapy and needle-free mesotherapy, respectively (Nature Cosmetics, Healthy Skin line, Siewierz, Poland), while group B had micro-needle mesotherapy with an ampoule with 0.9% of NaCl solution (Polpharma, Starogard Gdanski, Poland). Group B served as a control. The largest accumulation of elastic fibers in the dermis was observed in group C. A significant decrease in Ki-67 immunoexpression in the epidermis was observed in groups A and C by 57.8% and 55.4% [32]. An increased immunoexpression of the cellular proliferation marker Ki-67 is commonly reported in diseases based on the inflammation process (for instance, psoriasis) and pre-neoplastic lesions (for instance, actinic keratoses and cutaneous squamous cell carcinoma [32,64]. Marchlewicz et al. (2024) suggested that micro-needle mesotherapy improves skin condition on the basis of the thickness of collagen fiber bundles in the reticular layer of the skin, which increased by 43%, 33.8%, and 17% in groups A, B, and C, respectively [32]. Nevertheless, these results are considered to be difficult to interpret. On the one hand, the snail mucus mesotherapy resulted in decreased immunoexpression of the cellular proliferation marker Ki67 in the epidermis, which, according to the authors, can be connected to the regenerative properties of snail filtrate, but on the other hand, there was an increase in Ki67 and the proliferating cell nuclear antigen in the dermis. Ki67 is expressed in the basal layer of the epidermis and is strongly associated with tumor cell growth and proliferation. It is an essential enzyme for DNA replication, which plays an important role throughout the entire cell cycle with the exception of the M0 phase [64].

In one more study it was revealed that continuous use of a natural secretion of *C. aspersa* increased skin thickness with a simultaneous decrease in a mean epidermal thickness (like in publication [32]) and the signs of elastosis and an increase in the number of mean blood vessels per mm^2^, which, according to the authors, suggests enhanced skin cell proliferation and rearrangements of the fibrillar components of dermis and epidermis. Fifteen women aged 35–65 years applied an 8% emulsion of the natural secretion in the morning and a highly concentrated 40% liquid formulation at night daily for 3 months. However, these authors pointed out that large comparative and placebo-containing, double-blind, randomized studies were needed to establish the usefulness of these two products in the treatment of photoaging [60]. In a placebo-containing similar study, the daily application of the emulsion cream with 8% of natural secretion and 40% serum of natural secretion by 25 women improved periocular wrinkles and periocular and perioral texture after 12 weeks compared with an inactive placebo. However, the differences between the active and placebo sides were not great enough to be confirmed in the direct comparison test. Light cast in the parallel direction showed slight improvement in fine lines in both the preparations treated and placebo-treated skin [63]. To our mind, a slight difference can be explained by the presence of such biologically active substances as tocopheryl acetate, cetyl alcohol, and glycerin in the two placebos.

In addition, a randomized, double-blind, placebo-controlled trial [53] involved 22 participants aged 18–50 years with acne scars on both cheeks. This trial assessed the efficacy of the snail soothing and repairing cream, specifically developed to enhance post-procedural recovery following fractional laser ablation. The cream formula contained snail secretion filtrate, *Calendula officinalis* flower extract, tocopheryl acetate, *Glycyrrhiza glabra* root extract, *Aloe barbadensis* leaf juice, and many excipients. The placebo contained identical ingredients to the product, with the exception of snail secretion filtrate, which allowed establishing the effect of snail mucus. Each participant applied the cream and placebo on their assigned cheeks twice a day. The application of the cream contributed only to an increase in the degree of the stratum corneum hydration based on corneometry results; other clinical indications of skin damage recovery (edema, erythema, crust formation) did not differ from the placebo, and no significant side effects were observed. However, the authors indicated some limitations of their study. Among them are a limited number of participants and study visits, especially in the early post-laser period; relatively low interrater reliability in the physician rating scale; and skin biophysical measurements were not performed at the same time of day during each visit [53].

The study on the oil/water emulsion Apollinea Anti-Aging Face Cream, with a combination of *C. aspersum* snail mucus filtrate, donkey milk serum, sweet almond oil, and karite butter, showed a significant improvement in skin elasticity and hydration, as well as a reduction in skin wrinkles [54]. However, there was no placebo in this study. There was a limited number of participants (12 participants aged 35 and 67). The authors themselves indicated that all the components strongly contributed to the anti-aging efficacy of the final product. Therefore, it is impossible to conclude about the impact of snail mucus itself on the biological effect of the cream.

Based on the evaluation of skin hydration after the application of the moisturizing cosmetic creams on the basis of *Prunus amygdalus* dulcis oil and *Triticum vulgare* germ oil, which contained 0% (a control sample), 2%, 5%, and 10% of the *C. aspersum* mucus, the creams provided the desired moisturizing effect [55]. The authors also standardized the mucus according to the glycolic acid (3.35 g/L) and allantoin (0.41 g/L) contents. According to these values, three batches of the *C. aspersum* mucus are characterized by a rich content of glycolic acid and allantoin [33]. The results of corneometry showed that all the creams (2%, 5%, 10%) enhanced skin hydration compared to the cream without the snail mucus. The higher the concentration of snail mucus in the creams, the better the hydration of the skin in women [55]. Such results can be related to allantoin, an extremely moisturizing substance, and other components of snail mucus.

A randomized, double-blind, placebo-controlled efficacy and safety study of a combined serum was conducted in 66 subjects with mild to moderate acne. The serum contained 40% of snail secretion filtrate, 4% of extract of *Calendula officinalis*, and 0.3% of *Glycyrrhiza glabra* root extract (0.3%). The combination of these active ingredients is commercially available under the trade name of anti-inflammatory complex (AI complex^®^). The placebo had the same components except the anti-inflammatory complex. The product significantly reduced inflammatory acne lesions and had a favorable tolerability profile [56]. The authors explained these positive changes with the activity of three components, including several bioactive components of snail mucus, such as allantoin, glycolic acid, enzymes, glycosaminoglycans, proteoglycans, and achacin. To our mind, there are two drawbacks of this study. The first drawback of this study was the absence of a clear indication of which snail species had been used for the preparation of the facial serum. The second drawback is related to uncertainty about which component had an effect and to what extent. Wojnarowicz et al. (2021) [57] provided data from several studies in which, based on the subjective opinion of volunteers, the use of cosmetic products containing snail mucus resulted in an improvement in skin elasticity, hydration, and smoothness. The main drawback of these studies is the impossibility of determining the significance of each component of the cosmetic composition in improving the condition of the skin.

In recent years, a new trend has appeared in cosmetology–nutricosmetics that are used orally to ensure certain cosmetic effects. Bioactive peptides (2–20 amino acids long) are believed to play an important role in preventing skin anti-aging by improving the overall skin condition and providing protective effects, including inhibition of skin aging enzymes, antimicrobial, antioxidant, and anti-inflammatory effects [58]. In addition, they have fewer side effects as compared to synthetic products [59].

In vitro experiments, the water-soluble fraction of *C. aspersum* snail mucus showed promising inhibitory activity of thermal denaturation of human serum albumin, comparable to the effect of the well-known anti-inflammatory Diclofenac. The authors planned to develop a nutricosmetic product with snail mucus and comfrey root extract (*Symphytum officinale*) as the main bioactive ingredients, based on the significant content of allantoin, which has a significant antioxidant potential. However, it should be noted that the safety of *C. aspersum* snail mucus filtrate as a food additive has not been officially confirmed by the European state regulatory authorities.

Based on the analyzed papers, it is concluded that it is necessary to carry out placebo-controlled clinical trials in order to clearly distinguish the biological effects of a cosmetic form from its placebo. For the appropriate evaluation of biological effects, the researchers use two groups of participants, one group for the application of an investigational product and the second one for its placebo [32,56], or different parts of the body of the same participants [55], or two cheeks of the same participant for applying a product and its placebo [53].

## 6. Wound Healing Effect

Wounds, including burn wounds, are one of the most significant health issues in the world, which can lead to complications and even limb amputation. Microbial infections for burn wound patients are an extremely large problem because of a high level of mortality in hospitalized burn patients due to microbial infections [9]. In general, wound management remains a challenge in the clinic because of the high incidence of traumatic injuries and refractory chronic wounds [62].

Wound healing is a complex process that consists of four partially overlapping stages: hemostasis, inflammation, proliferation, and maturation or tissue modeling [62]. The efficiency of healing depends on the synchrony of the four phases and many internal and external factors. Moreover, different cytokines and growth factors control the wound-healing process. Among them are transforming growth factor-beta 1 (TGF-β1), which is synthesized by T cells, platelets, and macrophages, and vascular endothelial growth factor (VEGF), which is created by endothelial cells, fibroblasts, platelets, and neutrophils [9]. In addition, fibroblast migration and proliferation, macrophage differentiation, and neovascularization are observed during wound repair [27,62]. In general, fibroblasts secrete different cytokines involved in wound repair [27], collagen, and elastin [30].

The prospective use of snail mucus in medical practice for the treatment of wounds is actively discussed in the scientific community [9,27,62]. It was noted that the skin of workers became softer after contact with snail secretions during cleaning and manual handling of the animals. In addition, small cuts healed quickly, without infection or scarring [12,27,30,62,65]. It probably became the starting point for many studies related to the wound-healing effects of snail mucus.

Recent advances in molecular biotechnologies have made it possible to deepen the study of gastropod mucins as a scientific resource with a wide range of applications. Snail mucus of different snail species is proposed for the treatment of chronic lesions, skin conditions, post-surgical infections, stomach ulcers, etc. [62,65,66]. It is suggested that exogenous mucins can form a temporary protective layer over the epithelium, reducing the level of inflammation and leaving enough time for its regeneration. The hypothesis is based on the experiment findings showing that mice overexpressing mucin or mucin fragments showed a remarkable protection ability against pathogens in both the gut and the lungs. The exogenous delivery of mucins would certainly represent a simpler strategy compared to mucin overexpression. The barrier properties of mucins also suggest use for cell encapsulation and delivery. For example, a mucin capsule can protect probiotic bacteria against low pH and proteolytic activity in the stomach, so the bacteria enter the gut, where the local microflora can break down the mucins [65]. In addition, the unique chemical diversity and bioactivity of mucus determine their barrier properties, hydration, and lubrication properties, which are the basis for the development of new biomaterials—artificial tears and saliva, materials for drug delivery [62,65,66]. *C. aspersum* mucus agglutinin (lectin) is also considered a diagnostic tool that recognizes epitopes containing α-d-N-acetylgalactosamine present on the surface of metastatic cancer cells [67].

There are a lot of studies of *C. aspersum* mucus connected with the reparative properties of snail mucus [66,67,68,69]. Trapella et al. (2018) suppose that a synergistic activity of various molecules of snail mucus can induce the proliferation and migration of fibroblasts in in vitro wound repair. HelixComplex promoted cell migration and the wound-healing process by inducing the release from treated cells of IL-8, together with other still unidentified factors. IL-8 plays an important role in the reparative effects. A significant increase in the number of cells was also recorded in the mouse and human fibroblast cultures when exposed to snail mucus extract for 48 h and 72 h at a dose of 400 μg/mL compared to the untreated cells. Importantly, the effects on proliferation, migration, and wound healing remained unchanged after a long storage period (9 months) and several freezing (−80 °C)/thaw cycles, which indicated the stability of snail mucus, which is important for the pharmaceutical industry. A faster closure of the scratch and a significantly higher number of cells in the scratch area were observed in the cultures exposed to HelixComplex (400 μg/mL) with respect to control cultures [27]. An increase in normal human dermal fibroblast proliferation when exposed to snail mucus at a concentration of 15 μL/mL and 70 μL/mL was also observed in the MTT test by Polish researchers [30]. Similar studies were conducted by the other authors, using the scratch wound assay. The mucus of both snails improved the wound-healing process compared to untreated cells. The mucus of *E. desertorum* caused the migration of human skin fibroblasts, resulting in the complete wound closure after 48 h, which was faster compared to the mucus of *H. aspersa*. The gap made by the scratch was closed by 99.2% and about 80% after 48 h, respectively, compared to 55.1% in the control, after adding 400 μL of 300 μg/mL of the mucus of *E. desertorum* and *H. aspersa* [9]. Thus, the snail mucus increases cell migration, which is crucial for the wound-healing process.

The biological effects of the mucus extract were studied in vitro on the various skin cell populations: keratinocytes, macrophages, and fibroblasts [70,71,72,73]. An effect of a decrease in the number of alive cells (keratinocytes and macrophages) was revealed for the high concentration of the snail mucus (when diluted 1:20). The introduction of snail mucus into the fibroblast culture promoted its viability and effectively counteracted the basal expression of cyclooxygenase-2 (COX-2), depending on the concentration of the snail mucus. A significant reduction in COX-2 expression was revealed for human gingival fibroblasts compared to the control, with a major extent for 1:40 dilution, and the same tendency was observed with these cells pretreated with hydrogen peroxide 100 μM and then exposed to slime mucus at 1:40 and 1:80 dilutions. Moreover, snail mucus can lead to new blood vessel formation. Angiopoietin 1 gene expression is noticeably increased in the culture of human gingival fibroblasts when snail mucus was administered after the use of such an inflammatory stimulus as peroxyde hydrogen. Contrarily, the gene expression of the anti-angiogenic factor, angiopoietin 2, is reduced after adding snail mucus. Thus, snail mucus supported the positive modulation of the pro-angiogenic signaling [70]. These studies also allow reaching a conclusion that the concentration of snail mucus can have a great impact on wound healing, which is necessary to consider when developing medicinal products.

The efficacy of the purified *C. aspersum* mucus as a protective agent against ozone-damaging effects was confirmed in the experiments on human keratinocytes and reconstructed human epidermis models. In the test with the human keratinocytes, snail mucus in concentrations of 4–400 μg/mL did not demonstrate any toxic effects. The pretreatment with snail mucus improved the “in vitro” wound-healing process after 24 h compared to the control, in which only 50% of the wound area was recovered. According to the other experiment with the impact of ozone on the reconstructed human epidermis, a significant decrease was observed in the induction of proinflammatory cytokines (both IL-1beta and IL-8 mRNAs), and, at the same time, there was an increase in the anti-inflammatory mediator IL-10 after the pretreatment of the reconstructed human epidermis with snail mucus [47]. The effects of snail mucus enriched with antioxidant compounds obtained from *Acmella oleracea* L., *Centaurea cyanus* L., *Tagetes erecta* L., *Calendula officinalis* L., and *Moringa oleifera Lam* flowers were also studied on the human keratinocytes. UV radiation was used as a model for studying in vitro cytoprotective effects of the formulation, as UV radiation produces ROS in the skin. The results showed that irradiation and pretreatment and subsequent combined treatment with snail mucus and flower extract increased the content of glutathione and decreased the level of ROS and lipid peroxidation. Glutathione is essential for the maintenance of redox homeostasis. Therefore, restoration of its reduced form can counteract UV-induced oxidative stress [71]. The dried powdery extract of snail mucus caused a dose-dependent and time-dependent activation of keratocyte proliferation in the tested concentrations (0.01, 0.1, and 1 mg/mL). It increased proliferation at 72 h (80.15%, 70.63%, and 69.38%, respectively) compared to 48 h (73.11%, 61.31%, and 59.38%, respectively). It also showed pronounced antioxidant potential in the ORAC (oxygen radical absorbance capacity) test [72]. *C. aspersum* mucus antioxidant properties were confirmed in vitro by evaluating the total antioxidant capacity in the diphenyl-1-picrylhydrazine (DPPH) test and the iron reduction test. The results showed antioxidant activity with an IC50 of 58.45 ± 3.62 μg/mL in the DPPH test compared to ascorbic acid, which had an IC50 of 44.55 ± 0.61 μg/mL, as well as a strong iron reduction capacity (IC50 22.71 ± 0.67 μg/mL compared to ascorbic acid with an IC50 of 53.25 ± 3.94 μg/mL) [73]. Studies [71,72,73] could be a prerequisite for the development of sun protective products for skin care with the addition of snail mucus. In addition, the studies concerning angiogenesis [21,70,74,75,76] indicate that snail mucus has different mechanisms of impact on the wound-healing process. A new vascular system could deliver nutrients, oxygen, and various types of cells to the wound area [70]. Among these cells are inflammatory cells, fibroblasts, and mesenchymal cells, which secrete various growth factors. Mature endothelial cells can form new blood vessel walls in the wound [75,77]. The purified snail extract also induced angiogenesis in the transgenic Danio fish model, a useful experimental model for the study of the mechanisms controlling the formation of vasculature structures. The intensity of angiogenesis was evaluated according to the following parameters: formation of intersegmental vessels and subintestinal venous plexus, and modeling of the caudal venous plexus. Therefore, the mucus contains substances capable of increasing VEGF expression, which promotes angiogenesis [74]. VEGF is a marker of a proper wound-healing process [76]. A positive effect of snail mucus on the angiogenesis process was also reported in the experiments on male Wistar rats by Rosanto et al. (2021). Of the snail mucus, 24% did not significantly affect wound healing compared with the negative control, while 48% and 96% snail mucus showed substantial effects on angiogenesis in the papillary and reticular strata. The drawback of this study was the absence of a positive control [75]. A significant improvement was reported in the rate and percentage of model wound closure in experimental groups of the laboratory animals (mice and rats) vs. control with the topical application of filtered *C. aspersum* snail mucus, while wound-healing markers also improved (VEGF, alfa-sma (a marker of cell differentiation)) [21,76].

The rat excised wound models showed a synergistic effect of the gel on the base of *C. aspersum* mucus, *Plantago major* leaf extract, and *Calendula officinalis* flower extract vs. the untreated control and the gel compositions of *C. aspersum* mucus with only *P. major* leaf extract or *C. aspersum* mucus with only *Calendula officinalis* flower extract [77]. This paper also raises the issue of the average molecular mass of the mucus of *C. aspersum*. Kermedchiev et al. (2021) revealed that the mucus had substances with different molecular weights (25–35 kDa, 38–40 kDa, 45–50 kDa, 80–90 kDa, and above 250 kDa) detected with SDS-PAGE. Moreover, the mucus peptides with an MW below 3 kDa were determined by MALDI-MS analyses [77]. However, Cerullo et al. (2023) stated that the average molecular mass of proteins in the mucus secretions of *C. aspersum* was 30 kDa [31]. The drawback of the study [77] was the absence of tests with the gels on the basis of the individual components (*C. aspersum* mucus, or *Plantago major* leaf extract, or *Calendula officinalis* flower extract).

*C. aspersum* snail mucus showed good potential in bone and cartilage tissue regeneration in the composition of the chitosan matrix using highly proliferated Saos-2 and SW1353 cells [78,79,80]. In addition, the hydrogel biomaterial consisting of snail glycosaminoglycan and methacrylated gelatin, as well as pure mucus glycosaminoglycan, according to the histological and immunohistochemical findings, contributed to the decrease in skin inflammation and edema, the activation of angiogenesis during the treatment of model experimental diabetic wounds in laboratory animals [20,81].

The positive effect of nanogel containing *C. aspersum* mucus was reported in the treatment of hypertrophic scars in an experiment on laboratory animals [82]. In recent years, pilot studies have been carried out for the treatment or prevention of stomach ulcers by means of snail mucus. The gastroprotective effect of snail mucus filtrate was shown in the ethanol-induced and indomethacin gastric ulcer model in mice [29,83,84,85]. There is a clinical case report on treating the tongue traumatic injury with Gelenzimo^®^, which contained snail mucus and vegetable oils. The topical application of this preparation promoted the mucosal and submucosal healing process of the affected lingual site [68]. However, positive effects could be induced by the excipients of the preparation. The lack of clinical studies can be explained by the fact that the snail mucus extract is not listed as a medicinal substance, and therefore, its administration in medicine was not evaluated completely.

The performed critical analysis of the publications showed that snail mucus can influence different stages of the wound-healing process, including proliferation and angiogenesis; however, there is a lack of clinical studies proving these properties from in vitro and animal studies.

## 7. Antimicrobial Activity of *C. aspersum* Mucus

Given the potential use of snail mucus in cosmetic products and as an active agent of reparative processes in wound healing, the evaluation of its microbial purity and antimicrobial properties becomes particularly relevant. Moreover, nowadays, the number of pathogens resistant to antimicrobial medicinal products is rapidly increasing. For this reason, there is a need to search for novel molecules or products for overcoming drug resistance [5,7,21,22,69,78,86,87].

Overcoming microbial contamination of a native mucus allows obtaining a safe product with significant antibacterial properties. It is noted that *C. aspersum* mucus showed effective antibacterial activity against two different strains of *Pseudomonas aeruginosa* and *Streptococcus pyogenes*. Glycosylated peptides with a molecular mass of 4021.04 Da and 6403.73 Da, isolated from the mucus fraction of *C. aspersum* and *H. lucorum*, inhibited *Propionibacterium acnes*, *E. coli,* and *Helicobacter pylori* growth [78].

Most of the identified *C. aspersum* mucus peptides contain mainly Gly, Leu, Pro, Val, Phe, Ala, Asp, Asn, and Trp residues, which are characteristic of antimicrobial peptides (AMPs) and are important components of the innate immune system [5,22,77]. In general, AMPs are recognized as very advanced compounds with promising therapeutic potential and a wide spectrum of action against pathogenic microorganisms [5,8,77,88,89]. There are reports on developing effective wound dressings containing AMPs [90]. Most AMPs have a positive total charge and a hydrophobic surface. Both cationic and anionic peptides are present in mucus, but cationic peptides predominate and are characterized by an amphipathic structure and a predominantly hydrophobic surface. Their antimicrobial activity is connected to the ability to disrupt biological membranes as cationic peptides interact with anionic components of target cell membranes that can lead to membrane permeabilization, depolarization, leakage, or lysis, resulting in cell death. Peptides bind to the membrane surfaces with their hydrophobic sides anchored in the hydrophobic lipid core of the bilayer [5,88]. The risk of developing resistance to AMPs is low compared to conventional antibiotics. AMPs are believed to interact with the bacterial cell membrane through an electrostatic interaction, making it more difficult for bacteria to develop resistance, unlike conventional antibiotics. To date, there are several hypotheses regarding the mechanism of action and destruction of the microorganism membranes by AMPs, but this mechanism has not yet been fully established [8,10,91]. Lactic acid UCO-SMC3 strain bacteria were isolated from *C. aspersum* mucus, which is known for its in vitro ability to inhibit *Cutibacterium acnes* and *Staphylococcus aureus* [92].

It was revealed that the molecular mass of the fraction has an impact on the antibacterial selectivity. For instance, the fraction with a molecular weight of 10–30 kDa was active against *E. coli*. Another peptide fraction of a molecular weight of 20 kDa was active against the bacterial strain *Clostridium perfringens* [88]. The antimicrobial activity of the mucus fraction with an MW > 20 kDa was expressed against Gram-negative bacteria (*Salmonella enterica* 8691, *Enterococcus faecalis* 3915, and *Enterococcus faecium* 8754) even at concentrations of 8 μg/mL. At this concentration, more than 70% inhibition of bacterial growth was achieved [8].

The different concentrations of *H. aspersa* slime inhibited the growth of the Gram-positive and Gram-negative bacteria. The antibacterial activity depended on the concentration in the solution. A concentration of 100 μL/mL was the most effective concentration against *P. aeruginosa* isolated and identified at the Laboratory (PAAP lab), *L. monocytogenes* (ATCC 7644), *S. aureus* (ATCC 29213), and *E. coli* (ATCC 35218) [21]. Dolashki et al. (2018) identified peptides, which are supposed to have moderate antimicrobial activity. Fraction 1 (the compounds of molecular weight between 1 and 10 kDa) of the common brown garden snail *C. aspersum* in a concentration of 13.1 mg/mL showed the inhibition zones of 15 and 12 mm of *P. aeruginosa* AP9 and *Brevibacillus laterosporus* BT271. It was surprising that when this fraction was subdivided into two fractions, 1–3 kDa (a concentration of 6.9 mg/mL) and 3–10 kDa (a concentration of 5.3 mg/mL), antimicrobial activity was not revealed against these bacteria [22]. This fact could be explained by the dependence of antimicrobial activity on the concentration of a fraction or certain AMP.

Many studies have demonstrated the high antibacterial potential of mucus from various species of land snails. In particular, El-Zawawy and Mona (2021) [9] reported that mucus from the grape snail (*C. aspersum*) inhibits the growth of multidrug-resistant bacteria and fungi isolated from burn wounds, while mucus from the desert snail (*Eremina desertorum*) exhibits higher activity against these pathogens. Moreover, the antimicrobial activity depended on the concentration of mucus in a solution. Increasing the concentration from 10 to 50 μg/L caused an enhancement in the inhibition zones. It is worth noting that the snail mucus of *C. aspersum* inhibited the growth of some bacteria, including *P. aeruginosa* (PA-9) (15.8 mm), *E. coli* (EC-3) (16.8 mm), and *S. aureus* (SA-17) (18.2 mm) at a concentration of snail mucus of 50 μg/L. In addition, there was an inhibition of the growth of some fungi, including *A. niger* (AN-05) (8 mm), *Rhizopus stolonifer* (RS) (14 mm), *Trichoderma harzianum* (TH) (18 mm), and *Candida albicans* (CA-11) (13 mm). At the same time, mucus from both snails showed a pronounced anti-inflammatory effect and stimulated skin cell proliferation, indicating its wound-healing potential [9]. Similarly, a study by Aouji et al. (2023) [21] confirmed that the mucus of *H. aspersa* contains substances with antibacterial activity and significantly accelerates wound healing in model experiments.

The results of the study [8] confirmed the significant antibacterial potential of *C. aspersum* mucus, in particular its fractions with a molecular weight of <20 kDa and >20 kDa, which effectively inhibit the growth of five bacterial pathogens: *Bacillus cereus*, *P. acnes*, *S. enterica*, *E. faecalis*, and *E. faecium*. The analysis by de novo sequencing allowed the identification of 16 new peptides with antibacterial activity, while in the fraction > 20 kDa proteins, such as aspernin, hemocyanins, H-lectins, mucins, and L-amino acid oxidase-like protein, provided a synergistic effect. The obtained fractions demonstrated antibacterial activity at the concentrations of 32–128 μg/mL, which is comparable to the active substance. The MIC values demonstrated a higher antimicrobial potential of the fraction with an MW > 20 kDa of the mucus compared to the fraction with an MW < 20 kDa against *B. cereus* and *P. acnes*. Interestingly, the effective concentrations of mucus were comparable to the antibiotic vancomycin, while the mucus did not show cytotoxicity for eukaryotic cells. Additionally, antioxidant properties were found, which caused the reduction in ROS levels [8]. Proteins with molecular masses of 17.5 kDa and one of 18.6 kDa had activity against *Ps. aeruginosa* [93]. One more study also revealed antimicrobial proteins in the mucus of *H. aspersa*, which exhibit a strong activity against *P. aeruginosa* and a weak activity against *Staphylococcus aureus*. Antimicrobial activity against *P. aeruginosa* can be explained by one or more novel substances with molecular weights between 30 kDa and 100 kDa. Size separation experiments showed that the antimicrobial substances in *H. aspersa* were between 30 and 100 kDa, while electrophoresis showed one band was between 50 kDa and 60 kDa, the second was at approximately 35 kDa, and the other bands were noted at >260, 20, 15, 10, 12, and <10 kDa [69].

The antimicrobial and wound-healing properties of *H. aspersa* Müller slime could be used for future preclinical and clinical studies of creams, gels, and other dosage forms as an adjuvant therapy for skin wounds, considering the concentration, fraction of snail mucus, and its chemical composition in a dosage form.

## 8. Anti-Inflammatory Activity

The anti-inflammatory activity of snail mucus is studied on different models [9,12,27,37]. For instance, the anti-inflammatory activity of the mucus of both snails, *H. aspersa mucus* and *E. desertorum*, was determined in vitro through membrane stabilization of the human red blood cells, albumin denaturation, and proteinase inhibitory activity. Aspirin was used as a reference drug. Both snail mucus demonstrated anti-inflammatory activity, while *E. desertorum* showed higher activity [9]. Protein denaturation is one of the first effects of an inflammatory insult, triggering immune responses and oxidative stress [33].

In the studies with the proliferative canine keratinocyte cell line CPEK stimulated with LPS, it was revealed that snail mucus at the concentrations of 4% and 16% significantly diminished the release of the proinflammatory cytokines IL-6, IL-8, IL-17A, and TNF-α. IL-6 is a proinflammatory cytokine that plays a role in the increase and exacerbation of Th2-mediated diseases [12]. IL-8 stimulates the activation and recruitment of innate immune cells, such as neutrophils, at the site of inflammation; rouses proliferation, growth, and viability of vascular endothelial cells; and triggers cells by stimulating exocytosis and degranulation of storage proteins, particularly connected to the process of wound healing and inflammation [12,27]. IL-17 stimulates the production of antimicrobial factors and proinflammatory cytokines at the site of infection. Both inflammatory cytokines and endotoxins stimulate the production of COXs, key enzymes in inflammatory pathways [12,33]. It is worth mentioning that these effects and the viability of the cells in the presence of mucus depend on its concentration.

The novel lubricating multimolecular ophthalmological composition of snail mucus (GlicoPro-based artificial tears) demonstrated anti-inflammatory and analgesic activity, which could be used for the treatment of dry eye disease. This disease affects approximately 5–50% of the global adult population. The symptoms of this disease range from ocular discomfort to pain and vision disturbance. Hyperosmotic stress triggers conjunctival epithelial cells to release proinflammatory cytokines (interleukins IL-1α, IL-1β, IL-6) and chemokines (IL-8). These proinflammatory cytokines and chemokines drive a local inflammatory response, leading to ocular tissue damage and ocular surface pain. In addition, it was revealed that the conjunctival cell population in patients with dry eye disease had an increased percentage of ocular surface CD14+ (monocytes/macrophages) and neutrophils as a result of the proinflammatory cytokines and chemokines released by the conjunctival epithelial cells. The studies of GlicoPro^®^ were conducted on the human corneal epithelial cell line (HCE-2). The experiment findings showed a reduction in the level of inflammatory cytokines (IL-6, IL-8, IL-1β). Moreover, the sulfated mucopolysaccharide of *H. aspersa* mucus provided a desirable mucoadhesive strength for this ocular formulation compared with a 0.15% solution of sodium hyaluronate [37].

It is worth noting that the impact of snail mucus on cytokines depends on the type of cells and their concentration. In the cytokine analyses, fibroblast cultures were pre-exposed to the Helix Complex (400 μg/mL) or left untreated for 30 min, washed with PBS, and grown for an additional 24 h with normal fresh medium. Then, the culture supernatants were collected and analyzed for a panel of cytokines (IL-1α, IL-1β, IL-2, IL-4, IL-6, IL-7, IL-8, IL-10, IL-12, GM-CSF, IFN-γ, and TNF-α). The cytokine IL-8-specific release was significantly higher compared with the control supernatants [27].

The human keratinocyte cell line NCTC 2544 was stimulated with LPS, and the day after, treated with 10 mg/mL AuNPs-SME. In this study, AuNPs-SME reduced IL-6 and IL-8 proinflammatory cytokine transcription and increased anti-inflammatory IL-10 even at low concentrations (10 mg/mL). These results suggest that AuNPs-SME may have modulatory effects that could be beneficial in reducing inflammation [36].

Some formulations of snail mucus were tested: 40 wt% spray-dried slime, 60 wt% pectin/starch 1:1 (SS_SD_40), 20 wt% spray-dried slime, 80 wt% pectin/starch 1:1 (SS_SD_20), 100 wt% spray-dried slime (SS_SD_100), and Matrix_SD (0% snail mucus). All the samples significantly reduced IL-6 and IL-8 release from human colon adenocarcinoma-derived cells (Caco-2), cultured and differentiated to mimic enterocyte-like characteristic cells, compared to both LPS and Matrix_SD samples. What this experiment showed was that the SS_SD_40 sample significantly differed compared to the SS_SD_20 and SS_SD_100 samples for both IL-6 and IL-8, confirming once again the importance of snail mucus concentration in studies [33]. Cabibbo et al. (2025) considered such formulations with snail mucus as a first stage in the development of medicinal products for the treatment of gastric inflammatory diseases [33].

Therefore, regenerative properties and diminishing the level of expression of proinflammatory cytokines and COX-2 could be considered an attractive factor for the development of wound-healing, anti-inflammatory, and antiallergic creams and gels with snail mucus for the treatment of wounds and atopic dermatitis, respectively. Diminishing expression of proinflammatory cytokines, optimal mucoadhesive, and regenerative properties of snail mucus could be considered an appealing factor for the development of eye drops for the treatment of dry eye disease as well.

## 9. Other Effects

Given that oxidative stress, along with inflammatory and immune mechanisms, are critical factors in the pathogenesis of neurodegenerative diseases, and *C. aspersum* mucus has antioxidant potential, Bulgarian scientists investigated the effect of snail mucus extract on scopolamine-induced cognitive impairment and oxidative stress in the cerebral cortex of rats. According to the behavioral test results, a significant improvement effect was reported when the snail extract was administered *per os* on the learning and memory of animals treated with scopolamine [94]. The oxidative stress activation index also decreased and approximated the control group values. Later, the biochemical markers of neurodegenerative changes in the cerebral cortex and hippocampus were studied on a similar model: acetylcholinesterase activity, content of acetylcholine and monoamines (dopamine, norepinephrine, and serotonin), levels of primary oxidative stress markers, and expression of brain-derived neurotrophic factor (BDNF), cAMP response element-binding protein (CREB). Snail mucus extract was found to have inhibitory activity on acetylcholinesterase activity, moderate antioxidant properties, and the ability to modulate monoamine content in two brain structures [95]. Moreover, its long-term use not only recovered the scopolamine-suppressed CREB and BDNF expression but also significantly increased their regulation.

The application of snail mucins as antitumor agents is at the early stages of studies, but interest in this problem is constantly growing [63]. The protective effect of *C. aspersum* snail extract against damage caused by *N*-nitrosomethylurea was studied in mice. It is known that *N*-nitrosomethylurea causes changes in the hepatic and spleen circulatory system with hemangioma onset. *C. aspersum* extract significantly reduced the damage and preserved the histoarchitecture of these tissues [96].

*C. aspersum* extract was also found to induce necrosis of the Hs578T breast cancer cell line and significantly increase the expression of TNF-α. NF-κB and PTEN were inhibited at 1% dilution after 8 and 24 h of exposure [97]. An in vitro study on B16F10 cells and human melanoma IGR-39 and SK-MEL-28 cell lines showed that *C. aspersum* snail mucus has a potential anti-melanoma effect. Mucus specifically inhibited the viability of IGR-39 and SK-MEL-28 cells associated with the apoptotic effect but did not affect the viability of B16F10 cells and non-tumorigenic HaCaT cells [98]. Then, the inhibitory activity of *C. aspersum* mucus extract against human epithelioid carcinoma was found [99]. Domínguez-Martín et al. (2020) [100] report in their work that an aqueous extract of *C. aspersum* garden snail has antitumor effects on the necrosis-inducing breast cancer H5578T cell line; in addition, it is a potent stimulator of TNF-α and an inhibitor of NF-κβ, PTEN, and p53 factors that regulate tumor development.

Searching for new safe medicinal products to treat stomach ulcers is a topical issue, considering that peptic ulcer disease affects 4 million people each year worldwide [13,29]. Oxidative stress, inflammation, and neutrophil infiltration are involved in gastric ulcer pathogenesis and progression. ROS activate the nuclear factor kappa β (NF-κB) signaling pathway and facilitate the release of proinflammatory cytokines (interleukin 1β (IL-1β)), which worsen the gastric ulcer. Nuclear factor-erythroid 2-related factor 2 (Nrf2) and hemoxygenase-1 (HO-1) are related to the restoration of the antioxidant defense, suppressing NF-κB, thus inhibiting the signaling of proinflammatory cytokines. The preclinical studies were performed on eight groups of mice. The second, third, and fourth groups of mice received famotidine, low or high doses of mucin, respectively, without indomethacin. The stomach tissue contents of malondialdehyde, reduced glutathione, catalase, and NO showed no significant difference when compared with the normal control group (group 1). In the ulcerated control mice (group 5), gastric lipid peroxidation, expressed as malondialdehyde and NO, was significantly increased. The sixth, seventh, and eighth groups of the mice received famotidine (40 mg/kg b.w.), a low dose (7.5 mL/kg b.w.), or a high dose of mucin (15 mL/kg b.w.), respectively, before receiving indomethacin. These groups had significantly lower gastric mucosal malondialdehyde and NO contents, IL-1β and NF-κB expression, and increased gastric mucosal GSH and catalase contents, HO-1 and Nrf2 expressions, with regressions in gastric mucosal lesions compared to group 5 [29]. It is worth noting that similar results were revealed in the model of ethanol-induced ulcer. The intragastrical ethanol administration induced a significant increase in malondialdehyde levels and a decrease in catalase and SOD, while the pretreatment with omeprazole significantly prevented these changes. The treatment with snail mucus at the dose of 3 mL/kg did not show a significant protective effect, while the doses of 7.5 mL/kg and 15 mL/kg showed a significant protective effect in a dose-dependent manner [83]. It is assumed that this effect is due to the presence in snail mucus of molecules with antioxidant, tissue regeneration, and antimicrobial properties. For that reason, *C. aspersum* mucin could be regarded as a potential active substance of medicinal products for avoiding or treating gastric ulceration. In addition, there are studies that confirm a better local bioavailability of snail mucus in liposomes [101], which can be a strong pharmaceutical factor in the development of investigational medicinal products of snail mucus for topical application.

## 10. Conclusions

To summarize, this review demonstrates the regenerative, anti-inflammatory activity and antimicrobial properties of snail mucus formulations and discusses the practical value of *C. aspersum* snail mucus for cosmetology, dermatology, and other fields of medicine as a biologically active substance. Current data on the peptide composition of *C. aspersum* mucus provides a basic understanding of the bioactive component’s role in antimicrobial activity. Therefore, snail mucus could be regarded as a promising natural product due to its multifaceted biological properties. The biological effects of snail mucus are related to the combination of active molecules, including mucin, antimicrobial peptides, allantoin, and glycolic acid. In the future, there is a need to identify the molecular mechanisms underlying each of them. Studies that relate antimicrobial activity to the fraction of snail mucus continue, as antimicrobial activity of snail mucus depends on snail species, species of bacteria, concentration, fraction of snail mucus, etc.

Snail mucus can facilitate wound healing and has become an important subject in wound research. Experimental studies demonstrated that snail mucus’ wound-healing effects are connected to cell proliferation and migration, angiogenesis, and antimicrobial activity. Such properties might be the basis for the development of cosmetic and pharmaceutical products for the efficient induction of wound repair and pave the route for further studies of snail mucus as a natural product with regenerative properties. At this stage, there is a lack of safety and efficacy data on *C. aspersum* snail mucus products in clinical trials. We are aware that the findings from clinical randomized controlled trials are needed to confirm the therapeutic effects of snail mucus.

## Figures and Tables

**Table 1 molecules-30-04197-t001:** Characterization of snail mucus according to different authors.

Quality Index	Criteria of Acceptability				
	[33]	[27]	[37]	[29]	[28]
pH	4.5–6.5	6.0–7.0	7.0–8.0	4.80 ± 0.10	4.80
Density, g/mL	1.0–1.1	1.0–1.1	1.0–1.1	1.04 ± 0.01	1.02
Glycolic acid, g/L	>2	<0.2	<0.2	0.210 ± 0.00382	9.90
Allantoin, g/L	≥0.5	<0.02	<0.02	0.02	0.81
Dry residue after treatment at 105 °C for 3 h or Dry matter	0.5–2.0%	2–3%	0.1–0.2%		3%
Minerals, mg/L or Elements, mg/L	–	250–350	250–350	Calcium (1350 ± 3.05 mg/kg), potassium (1065 ± 2.88 mg/kg), phosphorus (955 ± 1.01 mg/kg), sodium (932 ± 2.52 mg/kg), magnesium (175 ± 2.52 mg/kg), iron (6.01 ± 0.03 mg/kg), Cr (0.009 ± 0.0005 mg/kg), Cu (5.09 ± 0.1 mg/kg), Hg (0.25 ± 0.01 μg/kg), Cd (0.019 ± 0.001 μg/kg), Co (0.001 ± 0.0006 μg/kg), Ni (0.95 ± 0.04 μg/kg), Zn (1.35 ± 0.02 mg/mL), Mn (0.69 ± 0.01 mg/mL)	Cr (0.007 mg/kg), Cu (5.04 mg/kg), Hg (0.22 m g/kg),
Vitamins				B_1_ (3.15 ± 0.005 μg/mL), B_2_ (0.75 ± 0.22 μg/mL), B_3_ (9 ± 1.53 μg/100 g), B_6_ (19.11 ± 0.11 μg/mL), B_12_ (9 ± 0.2 μg/100 g), A (0.99 ± 0.01 μg/mL), C (0.15 ± 0.006 mg/kg), E (0.11 ± 0.003 mg/kg)	
Polyphenols, mg/L	–	70–80	70–80	80 ± 1.51	–
Proteins, mg/L	–	100–250	90–200	250 ± 2.52	1540
Collagen, mg/L	–	1–100	–	85 ± 0.65	3200
Total bacterial count	<100 CFU/g	Absent	Absent	Absent	–
Yeasts and molds	≤10 CFU/g	Absent	Absent	Absent	–
Preservatives	–	–	Absent	–	–
Endotoxins, EU/mL	–	–	<0.25	–	–
Glycosaminoglycans sulfated, mg/L	–	29–90	29–90	–	–
Non-sulfated glycosaminoglycans (hyaluronic acid), mg/L	–	70–80	70–80	–	1000.0 (hyaluronic acid)

**Table 2 molecules-30-04197-t002:** Biological properties of the main components of snail mucus.

N	Component	Biological Properties	Reference
1	Collagen	It is a major component of the dermal extracellular matrix and the most common material used to enhance skin wound healing. Collagen is actively involved in various stages and pathways of regenerative processes, starting from angiogenesis to re-epithelialization. The success of this process depends on the coordination of growth factors, cytokines, and chemokines that affect cells through their binding to specific cell surface receptors or extracellular matrix proteins, of which collagen is the most common	[41]
2	Hyaluronic acid	It forms aggregates of proteoglycans. Its cross-linking with other matrix proteins, in particular collagen, leads to the formation of supramolecular structures. It also regulates cell migration and proliferation processes and creates a favorable environment for building the collagen-elastin skin matrix	[32,42]
3	Glycolic acid	It has the ability to penetrate the skin and increase collagen synthesis by fibroblasts. It can induce fibroblast proliferation at a concentration of 0.1 mM	[7,27,30,32,43]
4	Allantoin	It promotes cell proliferation and healing of the affected area of damaged skin, increases the water content in the extracellular matrix, and forms complexes with irritating and sensitizing substances. In addition, it facilitates the transition of a wound from an inflammatory to a proliferative state; it has direct antimicrobial effects	[30,32,34,36,44,45]
5	Vitamins A, E, and C	These vitamins exhibit antioxidant properties as they neutralize oxidants (free radicals, ROS, and nitrogen oxide), preventing the development of oxidative stress. Vitamin A accelerates cell renewal of the epidermis. The combined action of these vitamins positively affects the skin barrier functions—the moisture level of the stratum corneum recovers, keratosis reduces, which is clinically manifested in the form of small wrinkles, smoothing, leveling, and lightening of the basic skin tone	[46]
6	Polyphenols	They may partially determine its antioxidant properties, since their ability to neutralize pro-oxidant molecules is well known	[32,47]
7	AMPs and other substances	Antimicrobial properties	[8,9,22]

## Data Availability

No new data were created or analyzed in this study. Data sharing is not applicable to this article.

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
