# Peer review of "Chemical Variability and Biological Potential of *Cornu aspersum* Mucus as a Source for the Development of New Cosmetic and Pharmaceutical Products"

_molecules, 2025, doi:10.3390/molecules30214197_

Round 1

Reviewer 1 Report

Comments and Suggestions for Authors

The article “Chemical Variability and Biological Potential of Cornu aspersum Mucus as a Source for the Development of New Cosmetic and Pharmaceutical Products” tackles a relevant and timely subject, but its execution is problematic in several respects. Although the review assembles a large body of literature, the narrative lacks critical synthesis and instead becomes a long descriptive list of compounds, activities, and applications, often with substantial repetition. The discussion of mucins, allantoin, glycolic acid, and collagen, for example, reappears in multiple sections with little new insight, making the manuscript feel unfocused and poorly structured.

A fundamental weakness lies in the absence of a clear methodology for literature selection. The authors do not explain how studies were identified, which databases were searched, or how evidence was evaluated. As a result, weak, anecdotal, or preliminary findings are given the same weight as more rigorous investigations. In several cases, small, poorly designed clinical trials or even anecdotal observations are presented without critical appraisal, which risks overstating the strength of evidence. This problem is particularly evident in the sections on wound healing and cosmetic efficacy, where claims of positive effects are highlighted while study limitations are glossed over.

The issue of standardization, which the authors themselves identify as a major barrier for medical application of snail mucus, is treated superficially. While variability in chemical composition and microbiological contamination are mentioned, there is little attempt to compare existing standardization approaches, evaluate analytical methods, or suggest practical solutions. This omission is a missed opportunity, as one of the most urgent challenges in this field is precisely the lack of reproducible quality criteria.

The section on cosmetic applications is notably weak from a scientific standpoint. While the text cites market data and describes products, it does not offer a critical assessment of whether the reported cosmetic benefits are scientifically justified. Many of the cited cosmetic trials involve multiple active ingredients, making it impossible to isolate the role of snail mucus, yet this problem is barely addressed. The possibility of placebo effects and the lack of long-term safety evaluations are similarly overlooked, giving the section a promotional rather than an analytical tone.

The writing style also undermines the scientific quality of the article. Frequent use of vague phrases such as “it is believed,” “it is supposed,” or “some authors suggest” reduces precision and credibility. Digressions into historical anecdotes, snail farming, or market projections, while perhaps interesting, are presented at the expense of deeper mechanistic or critical analysis, further weakening the focus of the review.

Perhaps most concerning is the overly optimistic conclusion. Despite acknowledging the lack of randomized clinical trials and the unresolved difficulties in standardization, the authors still emphasize snail mucus as a promising therapeutic and cosmetic agent. This conclusion is not adequately supported by the evidence summarized in the review, which is highly heterogeneous, often preliminary, and in many cases contradictory. The article risks misleading readers by presenting speculative potential as if it were an established fact.

In summary, while the article compiles a wide range of information on Cornu aspersum mucus, it does not meet the standards of a critical scientific review. The main problems are lack of methodological rigor, unselective presentation of weak evidence, insufficient analysis of standardization challenges, superficial treatment of cosmetic applications, and conclusions that are far too optimistic given the data. The manuscript would benefit greatly from a systematic literature review approach, a structured and critical appraisal of evidence quality, and concrete proposals for addressing the barriers to clinical translation. Without these improvements, the work remains an extensive but uncritical survey rather than a meaningful contribution to the field.

Author Response

Reviewer #1:

Comments: The article “Chemical Variability and Biological Potential of Cornu aspersum Mucus as a Source for the Development of New Cosmetic and Pharmaceutical Products” tackles a relevant and timely subject, but its execution is problematic in several respects. Although the review assembles a large body of literature, the narrative lacks critical synthesis and instead becomes a long descriptive list of compounds, activities, and applications, often with substantial repetition. The discussion of mucins, allantoin, glycolic acid, and collagen, for example, reappears in multiple sections with little new insight, making the manuscript feel unfocused and poorly structured.

We agree with these remarks. We have gathered the table with the properties of the components. In such a way, we have outlined the information about the components and eliminated the repetitions in the manuscript. We hope that we have improved the readability of the manuscript.

A fundamental weakness lies in the absence of a clear methodology for literature selection. The authors do not explain how studies were identified, which databases were searched, or how evidence was evaluated. As a result, weak, anecdotal, or preliminary findings are given the same weight as more rigorous investigations. In several cases, small, poorly designed clinical trials or even anecdotal observations are presented without critical appraisal, which risks overstating the strength of evidence. This problem is particularly evident in the sections on wound healing and cosmetic efficacy, where claims of positive effects are highlighted while study limitations are glossed over.

Dear reviewer,  thank you for this remark. We eliminated anecdotal information, added methodology for literature selection and a lot of critical analysis.

The issue of standardization, which the authors themselves identify as a major barrier for medical application of snail mucus, is treated superficially. While variability in chemical composition and microbiological contamination are mentioned, there is little attempt to compare existing standardization approaches, evaluate analytical methods, or suggest practical solutions. This omission is a missed opportunity, as one of the most urgent challenges in this field is precisely the lack of reproducible quality criteria.

We added analytical methods used for standardization of snail mucus

The section on cosmetic applications is notably weak from a scientific standpoint. While the text cites market data and describes products, it does not offer a critical assessment of whether the reported cosmetic benefits are scientifically justified. Many of the cited cosmetic trials involve multiple active ingredients, making it impossible to isolate the role of snail mucus, yet this problem is barely addressed. The possibility of placebo effects and the lack of long-term safety evaluations are similarly overlooked, giving the section a promotional rather than an analytical tone.

We agree with this remark. Our aim was not to promote snail mucus. We added a lot of critical analysis, for example, that the placebo also can have biological effects due to the presence of substances with biological activity or there is a combination of preparations and it is difficult to conclude what acts.  . In general we added a lot of analytical tone. Therefore, we submit the revised version with delay.

The writing style also undermines the scientific quality of the article. Frequent use of vague phrases such as “it is believed,” “it is supposed,” or “some authors suggest” reduces precision and credibility. Digressions into historical anecdotes, snail farming, or market projections, while perhaps interesting, are presented at the expense of deeper mechanistic or critical analysis, further weakening the focus of the review.

We partly agree with your remarks. Why partly. According to the dictionary (https://dictionary.cambridge.org/grammar/british-grammar/be-expressions-be-able-to-be-due-to?q=Be+supposed+to) to be supposed is also to talk about people’s expectations or beliefs about something:[talking about some medicine]

We eliminated partly information about snail farming. We added a lot of critical analysis and speculation.

Perhaps most concerning is the overly optimistic conclusion. Despite acknowledging the lack of randomized clinical trials and the unresolved difficulties in standardization, the authors still emphasize snail mucus as a promising therapeutic and cosmetic agent. This conclusion is not adequately supported by the evidence summarized in the review, which is highly heterogeneous, often preliminary, and in many cases contradictory. The article risks misleading readers by presenting speculative potential as if it were an established fact.

We redid conclusions eliminating optimistic variants. A lot of publications, for instance, say about wound healing effects and improvement of the skin. We cannot ignore these facts. In fact, clinical trials are needed to confirm biological effects of snail mucus.

In summary, while the article compiles a wide range of information on Cornu aspersum mucus, it does not meet the standards of a critical scientific review. The main problems are lack of methodological rigor, unselective presentation of weak evidence, insufficient analysis of standardization challenges, superficial treatment of cosmetic applications, and conclusions that are far too optimistic given the data. The manuscript would benefit greatly from a systematic literature review approach, a structured and critical appraisal of evidence quality, and concrete proposals for addressing the barriers to clinical translation. Without these improvements, the work remains an extensive but uncritical survey rather than a meaningful contribution to the field.

Response. We appreciate this critical point. As was said we added a lot of critical analysis.

We have tried to take into account all your critical comments and have made significant changes to the text of the revised manuscript. We believe that taking these points into account will significantly improve the clarity, accuracy, and impact of our review. We also hope that our corrections will improve the readability of the manuscript. Thank you for your recommendations.

Reviewer 2 Report

Comments and Suggestions for Authors

The paper by Hudz et.al. describe the biological properties of the snail mucus from Cornum aspersum (Helix aspersa) species. The paper well describes the different application, starting form cosmetics till the more interesting pharmaceutical activity of this complex mixture of compounds.

Only few suggestion for the authors:

1) In the antibacterial activity of the mucus I think the authors missed the following literature evidence: http://dx.doi.org/10.1080/09674845.2016.1155377, in which is, for the first time, described a small protein with antibacterial activity.

2) The following paper could be added to the literature review in which the snail mucus is formulated in a liposome nanostructure for pharmaceutical purpose: https://doi.org/10.3390/molecules26164709.

Minor concerns:

carefully read trogouth the text the references that in some times are not in bold, for example page 9, line 376 …Skin woman [56]. [56] must be in bold. The same are missed along the text sometimes.

I suggest to publish the paper after this minor suggestion.

Author Response

Reviewer #2:

The paper by Hudz et.al. describe the biological properties of the snail mucus from Cornum aspersum (Helix aspersa) species. The paper well describes the different application, starting form cosmetics till the more interesting pharmaceutical activity of this complex mixture of compounds.

Only few suggestion for the authors:

We are very thankful for your valuable recommendations. 

Comment 1): In the antibacterial activity of the mucus I think the authors missed the following literature evidence: http://dx.doi.org/10.1080/09674845.2016.1155377, in which is, for the first time, described a small protein with antibacterial activity.

Response. We thank the reviewer for this helpful suggestion, and added this literature to the References and text of the manuscript.

In fact, we added this paper. Antimicrobial properties of mucus from the brown garden snail Helix aspersa

S J Pitt, M A Graham, C G Dedi, P M Taylor-Harris, A Gunn DOI: 10.1080/09674845.2015.11665749

It is very valuable; it shows antimicrobial properties of snail mucus depending on the fraction.

http://dx.doi.org/10.1080/09674845.2016.1155377 is Letter to the Editor: Antimicrobial properties of mucus from the brown garden snail Helix aspersa

Comment 2): The following paper could be added to the literature review in which the snail mucus is formulated in a liposome nanostructure for pharmaceutical purpose: https://doi.org/10.3390/molecules26164709.

Response. Thank you. Done.

Minor concerns:

Comment 3): carefully read trogouth the text the references that in some times are not in bold, for example page 9, line 376 …Skin woman [56]. [56] must be in bold. The same are missed along the text sometimes.

Response. We thank the reviewer for the attentiveness, and have corrected the text of the manuscript.

Reviewer 3 Report

Comments and Suggestions for Authors

In the review titled "Chemical Variability and Biological Potential of Cornu aspersum Mucus as a Source for the Development of New Cosmetic and Pharmaceutical Products," Nataliia Hudz et al. summarize the current literature on the use of C. aspersum mucus in both cosmetic and medical fields, considering the various types of activity exhibited by this product. They conclude that "snail mucus's biological effects pave the way for further studies of its potential as a cosmetic or pharmaceutical agent, including its standardization, nonclinical, and clinical studies." In my opinion, while the review is interesting in light of the potential for reducing the use of synthetic chemicals, it requires significant revision.

Overall, the introduction is very confusing and the paragraph structure is inconclusive. In my opinion, the paragraphs into which it is divided do not belong in an introduction. I would propose a general introduction, adding the purpose of the review, and then group the various types of activities into paragraphs with indicative titles. I would also add explanatory tables from time to time.

Below are some further indications

• English needs improvement

• Improve Table 1

• Standardize the bibliographical information

• When the text says: Author et al., it should always be in bold and the year of publication should always be added in parentheses

• Species names should always be in italics

• Next to Muller, the date the name was assigned should be put in parentheses

• Check for typos and spaces

• In the title BIOLOGICAL POTENTIAL, use normal text

• Lines 352-358 are just a very long list of molecules; indicate the essential information related to what you mean.

• Line 381: be careful of repetitions

Comments on the Quality of English Language

English needs to be improved

Author Response

Reviewer #3:

In the review titled "Chemical Variability and Biological Potential of Cornu aspersum Mucus as a Source for the Development of New Cosmetic and Pharmaceutical Products," Nataliia Hudz et al. summarize the current literature on the use of C. Aspersum mucus in both cosmetic and medical fields, considering the various types of activity exhibited by this product. They conclude that "snail mucus's biological effects pave the way for further studies of its potential as a cosmetic or pharmaceutical agent, including its standardization, nonclinical, and clinical studies." In my opinion, while the review is interesting in light of the potential for reducing the use of synthetic chemicals, it requires significant revision.

Comment: Overall, the introduction is very confusing and the paragraph structure is inconclusive. In my opinion, the paragraphs into which it is divided do not belong in an introduction. I would propose a general introduction, adding the purpose of the review, and then group the various types of activities into paragraphs with indicative titles. I would also add explanatory tables from time to time.

Response. Thank you for the comment. Done.

Below are some further indications

Comment: English needs improvement

Response. Done.

Comment: Improve Table 1

Response. Done.

Comment: Standardize the bibliographical information

Response. Done.

Comment: When the text says: Author et al., it should always be in bold and the year of publication should always be added in parentheses

Response. Done.

Comment: Species names should always be in italics

Response. Done.

Comment: Next to Muller, the date the name was assigned should be put in parentheses

Response. Done.

Comment: Check for typos and spaces

Response. Thank you for the comment. Done.

Comment: In the title BIOLOGICAL POTENTIAL, use normal text

Response. Done.

Comment: Lines 352-358 are just a very long list of molecules; indicate the essential information related to what you mean.

Response. Thank you for the comment. Done.Comment: Line 381: be careful of repetitions

Response. Sorry, there are no repetitions. Please specify.

In general thank you for this remark, we eliminated a lot of repetitions in the manuscript. We hope we significantly improve our manuscript.

Round 2

Reviewer 3 Report

Comments and Suggestions for Authors

The authors have reviewed the paper as requested, so it can be published in Molecules

Author Response

Dear reviewer, thank you very much for your work with our paper.